# Metabolic syndrome predictive modelling in Bangladesh applying machine learning approach

Md Farhad Hossain[ID]¹,², Shaheed Hossain²*, Mst. Nira Akter², Ainur Nahar², Bowen Liu[ID]¹, Md Omar Faruque³

1 Division of Computing, Analytics and Mathematics, Department of Mathematics and Statistics, School of Science and Engineering, University of Missouri, Kansas City, MO, United States of America, 2 Department of Statistics, Comilla University, Cumilla, Bangladesh, 3 Division of Energy, Matter and Sciences, School of Science and Engineering, University of Missouri, Kansas City, MO, United States of America

* shaheedhossaincou98@stud.cou.ac.bd

**Data Availability Statement:** Data set is publicly available in https://datadryad.org/stash/dataset/doi:10.5061/dryad.zkh18937f.

## Abstract

Metabolic syndrome (MetS) is a cluster of interconnected metabolic risk factors, including abdominal obesity, high blood pressure, and elevated fasting blood glucose levels, that result in an increased risk of heart disease and stroke. In this research, we aim to identify the risk factors that have an impact on MetS in the Bangladeshi population. Subsequently, we intend to construct predictive machine learning (ML) models and ultimately, assess the accuracy and reliability of these models. In this particular study, we utilized the ATP III criteria as the basis for evaluating various health parameters from a dataset comprising 8185 participants in Bangladesh. After employing multiple ML algorithms, we identified that 27.8% of the population exhibited a prevalence of MetS. The prevalence of MetS was higher among females, accounting for 58.3% of the cases, compared to males with a prevalence of 41.7%. Initially, we identified the crucial variables using Chi-Square and Random Forest techniques. Subsequently, the obtained optimal variables are employed to train various models including Decision Trees, Random Forests, Support Vector Machines, Extreme Gradient Boosting, K-nearest neighbors, and Logistic Regression. Particularly we employed the ATP III criteria, which utilizes the Waist-to-Height Ratio (WHtR) as an anthropometric index for diagnosing abdominal obesity. Our analysis indicated that Age, SBP, WHtR, FBG, WC, DBP, marital status, HC, TGs, and smoking emerged as the most significant factors when using Chi-Square and Random Forest analyses. However, further investigation is necessary to evaluate its precision as a classification tool and to improve the accuracy of all classifiers for MetS prediction.

## Introduction

Metabolic syndrome (MetS) is a cluster of conditions that, when combined, increase the vulnerability of a patient to several life-threatening diseases such as coronary heart disease, diabetes, stroke, and various other consequential health issues [1]. Approximately 422 million

**Funding:** The author(s) received no specific funding for this work.

**Competing interests:** There is no conflict of interest among the authors.

individuals globally suffer from diabetes, with the majority residing in low and middle-income nations [2]. The disease is directly responsible for 1.5 million fatalities annually. Over the past few decades, there has been a steady rise in both the number of cases and the incidence of diabetes. Metabolic syndrome, also known as insulin resistance syndrome and syndrome X, encompasses a range of risk factors related to blood pressure, glucose, and plasma lipid levels. Hypertension was identified by a systolic blood pressure of at least 130 mm Hg, a diastolic blood pressure of at least 85 mm Hg, or the use of antihypertensive medications. The risk of sudden cardiac death was found to be 70% higher in those with the MetS. According to the United States National Heart Lung and Blood Institute, every 1 in 3 Americans is suffering from MetS [3]. A 2018 non-communicable disease risk factor evaluation revealed that 15.5% of Bangladeshis aged 40–69 are at risk for cardiovascular diseases [4]. According to a systematic review and meta-analysis [5], 37.0% of Bangladeshi people were found to have MetS. The investigation of potential risk factors is necessary due to the high prevalence of MetS, which is a serious public health concern. According to the National Institute of Health, with advancing years comes an increased risk of MetS [6]. The risk of MetS may increase due to lifestyle choices, being inactive, poor diet, lack of quality sleep, smoking, excessive alcohol consumption, poor socioeconomic standing, and working irregular shifts. The chances of MetS in adults with a sleeping duration of less than six hours per day were roughly five times higher than the odds of MetS in adults who slept often and for more hours per day [OR: 4.62; 95% CI: (1.02, 20.98)] [7]. Genetic and family history and obesity also worsen the condition as these conditions can reduce the "good" HDL cholesterol and increase the "bad" LDL cholesterol while impacting badly on blood triglycerides, and blood pressure [8]. Additionally, certain medical conditions such as polycystic ovary syndrome (PCOS) and insulin resistance can also increase the risk of developing MetS [9]. Pregnancy-related overweight and obesity can increase the child's chance of developing MetS [10]. The impacts of socioeconomic conditions on health are demonstrated in a study where they showed that rural populations have significantly higher tobacco use (45.2%), inadequate fruit/vegetable intake (92.1%), and higher daily salt intake (9.0 g) compared to urban populations [11]. Socioeconomic condition, age, sex, obesity, hypertension, wealth, and living conditions all had impacts on the prevalence of diabetes according to the Bangladesh Demographic and Health Survey 2017–18 [12]. Of those with diabetes, 61.5% were not reported of the condition, 35.2% were receiving regular treatment, and 30.4% had it under control [13]. Predictive healthcare strategies, utilizing machine learning (ML), are crucial in predicting and mitigating the impact of metabolic diseases, particularly in countries like Bangladesh with rising prevalence of diabetes, obesity, and cardiovascular diseases. ML significantly enhances data-driven research efficiency, reducing manual inspection burden and enabling the development of new models for optimal operation. ML uses feature selection techniques which can be carried out via random forests, chi-square tests, and correlation plots to identify key factors in large datasets with numerous less significant variables. Our study aims to identify the potential risk factors associated with metabolic diseases and to generate a predictive model of such diseases using anML approach. ML is preferred for accurate prediction due to its ability to recognize patterns and relationships, analyze larger data sets, and generate predictions quickly, making it more efficient than traditional methods, which are time-consuming and vulnerable to bias [14]. According to a nationwide cross-sectional survey carried out in 2018, 12.3% of adult Bangladeshis reported engaging in insufficient physical activity, with women (14.8%) and urban groups (14.1%) exhibiting higher frequencies. Additionally, the study found that the prevalence of overweight and obesity was 25.9% and that it was considerably higher in the groups of women (33.7%), urban (34.3%) and wealthiest (34.3%) [11].

In this research, we aim to identify the risk factors that have an impact on MetS. Subsequently, we intend to construct predictive ML models and ultimately, assess the accuracy and reliability of these predictive ML models.

## Related work

In the last 30 years, there has been a significant rise in the global prevalence of MetS [15]. The International Diabetes Federation (IDF), the World Health Organization (WHO), the European Group for the Study of Insulin Resistance (EGIR), and the US National Cholesterol Education Program Adult Treatment Panel III (NCEP ATP III) have defined and published separate clinical criteria for MetS [16].

Mohammad Ziaul Islam Chowdhury et al. proposed an examination of the MetS prevalence in Bangladesh using meta-analysis [5]. The findings indicated that the prevalence of MetS in females (32%) is higher than in males (25%), although this difference is not statistically significant (p = 0.434). When the modified NCEP III criteria were utilized, the highest occurrence of metabolic syndrome was observed at 37%. Conversely, the prevalence decreased to its lowest level of 20% when the WHO criteria were applied. The studies indicated that geographical factors played a significant role in the variation observed [5].

Another method proposed by Suparno Datta et al. utilized ML to detect MetS at an early stage [15], which relies exclusively on non-invasive features such as height, weight, waist circumference (WC), triglycerides (TGs), blood sugar, and HDL levels. The ensemble learning approach demonstrates superior performance, with GBMs and RF closely trailing behind. The findings indicate that machine learning can effectively forecast MetS, eliminating the need for invasive biomarkers, and enhancing the convenience of early detection [15].

Guadalupe Obdulia Gutiérrez-Esparza et al. proposed an ML algorithm that predicts MetS in the Mexican population [17]. Random Forest was employed to prioritize health parameters. The key prognostic factors for MetS, based on their significance, included FPG, TGs, WHtR, HDL-C, and BMI. The data was analyzed using the Random Forest and chi-squared methods, which showed that WHtR had the highest values and was the most significant factor. Additionally, when evaluated with the C45 and JRip algorithms, WHtR demonstrated better performance in terms of balanced accuracy, sensitivity, and specificity. The RF model, which utilized ATP III variables, demonstrated the highest performance with a balanced accuracy of 0.875, closely trailed by JRip [17].

Shu-Jie-Xia et al. developed a diagnostic model for MetS that can be developed by incorporating symptoms into a physiochemical index [18]. Their selected cohort was compared to three traditional machine learning methods: Decision tree (DT), Support vector machine (SVM), and Random Forest (RF). Comparison among the three models indicated that the RF model exhibited superior performance, boasting the highest average accuracy (0.942 on average, with a 95% CI of [0.925, 0.958]) and sensitivity (0.993 on average, with a 95% CI of [0.990, 0.996]), when compared to SVM. The significance of the TGM indexes in predicting MetS was clearly stated in this study [18].

DarkoIvanovic et al. proposed an Artificial Neural Network (ANN) that can be utilized to predict the diagnosis of MetS by solely relying on non-invasive, cost-effective, and readily available diagnostic methods [16]. They included gender, age, body mass index (BMI), waist-to-height ratio, and systolic and diastolic blood pressure as the input vectors. The outcome of this study demonstrated that the implementation of ANN effectively predicts both positive and negative cases of MetS, thereby aiding in the early prevention of metabolic syndrome. The highest positive predictive value (PPV) was found to be 0.858, while the negative predictive value (NPV) was close to PPV at 0.832 [16].

Hui Zhang et al. also used ML in a retrospective cohort study to predict the probability of adults developing MetS within a 4-year period. Three ML techniques were selected, namely ANN, classification, regression tree, and SVM. All models, except for the classification and regression tree model in internal validation, had discrimination values greater than 0.7. In external validation, the Logistic regression model showed the highest discrimination. Furthermore, both external validation (0.780) and internal validation (0.788) demonstrated satisfactory calibration for the ANN model [19].

Mohammad Salim Hossain et al. examined the MetS among individuals with diabetes who reside in aBangladeshi coastal area [20]. It was discovered that approximately 47.00% of patients diagnosed with type 2 diabetes mellitus were afflicted with MetS. The prevalence rate of MetS was higher in females, with 58.60%, compared to males, who had a rate of 36.14%. Females showed higher rates of obesity and hypertriglyceridemia, along with lower levels of HDL. Also, the age group of 55–64 showed the highest occurrence of MetS [20].

Suresh Mehata et al. evaluated the occurrence and factors influencing MetS in Nepalese adults based on a study that represents the entire nation. The most common combination was low HDL-C, abdominal obesity, and high blood pressure, making up 8.18% of cases. Close behind was abdominal obesity, low HDL-C, and high triglyceride levels, accounting for 8% of cases. Only a small fraction, specifically less than two percent, of the participants exhibited all five components of the syndrome, while a significant portion, 19%, did not display any of the components. The prevalence of the syndrome consistently increased as the age group advanced, with adults between the ages of 45 and 69 having the highest prevalence, ranging from 28% to 30% [21].

Hayat Ali Shah et al. used deep neural networks to generate feature representations of metabolic pathways, which are then fed into random forests for pathway prediction. The DeepRF model accurately predicts both known and unknown metabolic pathways in organisms. It has been tested on a dataset of over 318,016 instances, showing high accuracy (>97%), recall (>95%), and precision (>99%). When compared to other methods, DeepRF consistently provides more reliable results [22].

## Methodology

### Materials

**Data source.** The dataset in this research was taken from the survey "National STEPS Survey for Non-communicable Diseases [NCDs] Risk Factors in Bangladesh 2018" which was conducted by the National Institute of Preventive and Social Medicine (NIPSOM) under the World Health Organization (WHO) [11]. Dataset link (STATA): https://extranet.who.int/ncdsmicrodata/index.php/access_licensed/download/1763/5374,(CSV):https://extranet.who.int/ncdsmicrodata/index.php/access_licensed/download/1763/5375. The study involved 8185 respondents, with 3804 male (46.5%) and 4381 female (53.5%) participants aged between 18–69 years. The dataset encompasses various risk factors for metabolic diseases including lifestyle habits, clinical and anthropometric measurements, and biomedical evaluation. A national cross-sectional population-based survey utilized a multi-stage cluster sampling design to select households and eligible adult men and women (aged 18–69) for an interview and physical examination. The physical examination consisted of anthropometry, blood pressure measurement, blood glucose, cholesterol, and a urine sample for salt analysis. The WHO NCD STEPS instrument version 3.2 was utilized to carry out the survey. The questionnaire is comprised of three STEPS aimed at assessing the NCD risk factors. Each step encompassed a range of core, expanded, and country-specific questions that were adjusted to cater to the local requirements. In Bangladesh, all core modules and optional modules, namely oral health, and cervical cancer

screening, were included. The questionnaire was translated into Bengali, and the validation of the translated questionnaire was conducted through back translation. In the initial phase, personal information from participants in STEP 1, which included documenting their height, weight, and hip and waist measurements was collected. These measurements were taken from individuals who agreed to move on to STEP 2. After completing data collection in STEP 1 and STEP 2 at selected households, the following day, biochemical assessments were carried out at specified locations for each Primary Sampling Unit (PSU). These assessments involved analyzing blood samples for glucose and total cholesterol levels, which were taken from venous blood samples. Plasma samples were also used to measure the concentrations of glucose, total cholesterol, and HDL cholesterol. Fasting blood samples were specifically collected to identify elevated blood glucose levels. The subjects were classified as having type 2 diabetes if they reported being informed by their doctor about the disease (provided the diagnosis was made after the age of 25 and not due to pregnancy), if they reported using insulin or a hypoglycemic medication, or if their fasting blood sugar level exceeded 100 mg/dL [23].

**Habits and lifestyles.** Three STEPS were used in validated questionnaires [11] to measure the risk factors for NCDs. These questionnaires were used to collect data on lifestyle variables such as alcohol and smoking consumption, physical activity levels, and salt intake before meals.

**Clinical and anthropometric measurements.** The measurements of the diastolic and systolic blood pressure were taken following the JNC-established standard protocol [24], WC, height, weight, and BMI were determined using the formula weight/height$^2$. The WHtR was computed by dividing the WC by height (waist/height).

**Biochemical evaluation.** The laboratory tests that were acquired were fasting blood glucose (FBG),TGs, and HDL cholesterol (HDL-C). Blood samples were obtained after a 12-hour overnight fast.

**Diagnostic criteria.** MetS encompasses a combination of significant risk factors, lifestyle-related risk factors, and emerging risk factors. These factors include abdominal obesity, atherogenic dyslipidemia (elevated triglyceride levels, small LDL particles, low HDL cholesterol), high blood pressure, insulin resistance (with or without glucose intolerance), and prothrombotic and proinflammatory states [25]. The clinical diagnosis criteria for MetS based on the ATP III criteria [26] was used. It is shown in Table 1.

## Methods

**Decision tree.** A decision tree is a tree-structured classifier in which the features of a dataset are represented by internal nodes, the decision rules are represented by branches, and the conclusion is represented by each leaf node. It is a graphical tool that shows all the options for

**Table 1. The clinical diagnosis criteria for metabolic syndrome based on the ATP III criteria.**

| Risk Factors | Criteria |
|---|---|
| MetS | Three or more of the following criteria |
| Waist Circumference | Male: >102 cm, Female: >88 cm |
| Systolic Blood Pressure | ≥130 mmHg |
| Diastolic Blood Pressure | ≥85 mmHg |
| TGs | ≥150 mg/dL |
| Fasting Blood Glucose | ≥100 mg/dL |
| Body Mass Index | ≥30 kg/$m^2$ |
| HDL | <40 mg/dL |

solving a problem or making a decision given certain parameters [27]. The method is non-parametric, effective for large datasets, and can be divided into training and validation datasets for optimal decision tree model construction [28].

**Random forest.** Adele Cutler and Breiman [29] introduced Random Forest which is a prediction technique that generates a set of CART classification trees and assigns the class to the instance based on a majority vote. This approach outperforms individual classification trees in terms of prediction accuracy and can be used for a variety of prediction situations [30]. In cases of regression or classification, Random Forest offers a technique called Variable Importance Measures (VIMs) to rank the importance of variables.

**Support vector machine (SVM).** A support vector machine (SVM) is an ML algorithm that uses supervised learning models to solve complex classification, regression, and outlier detection problems by performing optimal data transformations that determine boundaries between data points based on predefined classes, labels, or outputs. SVMs are widely adopted across disciplines such as healthcare, natural language processing, signal processing applications, and speech & image recognition fields [31].

**Extreme gradient boosting (XGBoost).** An ensemble ML technique that uses decision trees to provide a gradient boosting framework is called Xtreme Gradient Boosting (XGBoost). To reach the final prediction, XGBoost creates new models that predict the residuals of the earlier models [32].

**K-nearest neighbors (KNN).** The k-nearest neighbors technique transforms Big Data into Smart Data, free from noise, redundant information, and missing values [33]. This approach is crucial for accurate data mining and revealing insightful information.

**Logistic regression.** A logistic regression model examines the relationship between one or more independent variables that are already present in order to predict a dependent variable in the data. Multiple input criteria can be considered by the model. Logistic regression is used in the field of ML as a key technique. The algorithms improve at classifying data sets as more pertinent data becomes available.

**Feature selection criteria.** The process of feature selection holds great significance in the realm of ML model development it allows the identification of crucial variables from extensive datasets which have a significant influence on the model in comparison to other variables. In our study, we utilized chi-square and random forest methodologies to identify these important variables.

**Statistical analysis.** To reveal the characteristics of objects under study we exert a chi-square test employing statistical packages for social science by using SPSS software version 28.0.We also utilized Python 3.0 with Jupyter Notebook where Pandas, NumPy, Scikit-learn, Seaborn, and Matplotlib libraries were used to reveal the results.

## Metrics

In ML, various performance metrics are used to evaluate the accuracy and effectiveness of a classification model. Some of the key metrics include:

**Precision:** The proportion of true positive predictions among all positive predictions made by the model. It measures how accurate the model's positive predictions are [34].

**Sensitivity:** Also known as recall, it is the proportion of true positive predictions among all actual positive cases. It measures how well the model identifies all the positive cases [34].

$$\text{SENS} = \frac{TP}{TP + FN}$$

**Specificity:** It is the proportion of true negative predictions among all actual negative cases.

Specificity measures how well the model identifies all the negative cases [34]salma.

$$SPC = \frac{\text{TN}}{\text{FP} + \text{TN}}$$

**Balanced accuracy (AUC-ROC):** The area under the receiver operating characteristic (ROC) curve, which measures the model's ability to distinguish between positive and negative cases. A higher AUC-ROC value indicates a better model performance [35].

$$BACC = \left(\frac{1}{2}\right)\left(\frac{TP}{P} + \frac{TN}{N}\right)$$

where P = Positive, N = Negative, TP = True Positive, FN = False Negative, TN = True Negative and FP = False Positive, respectively.

**F1 score:** The harmonic means of precision and recall; it measures the model's overall accuracy by balancing both precision and recall. A higher F1 score indicates better model performance [36].

$$F1 = \frac{2 \times precision \times recall}{precision + recall}$$

## Results and analysis

In our research, we utilized the dataset obtained from the "National STEPS Survey for Noncommunicable Diseases Risk Factors in Bangladesh 2018" [11]. The ATP III criteria were employed to identify the crucial cardiovascular risk factors, and subsequently, the participants were categorized into two groups: MetS Group and Normal Group. The variables in our study can be classified into four categories: Lifestyle variables, Anthropometric variables, Clinical variables, and Biochemical variables [26, 37]. Lifestyle variables encompass factors such as smoking, alcohol consumption, and marital status. Anthropometric variables include age, weight, height, BMI, WHtR, waist circumference (WC), and hip circumference (HC). Clinical variables consist of systolic and diastolic blood pressure measurements. Lastly, biochemical variables encompass fasting blood glucose (FBG), high-density lipoprotein (HDL), and triglycerides (TGs).

In Fig 1, the process of constructing our proposed model is depicted. Initially, the crucial variables are identified through the utilization of Chi-Square and Random Forest techniques [17]. Subsequently, the obtained optimal variables are employed to train various models including Decision Trees, Random Forests, Support Vector Machines, Extreme Gradient Boosting, K-nearest neighbors, and Logistic Regression. The validity of these models is then assessed, and their performance is compared based on metrics such as Precision, Sensitivity, Specificity, Balanced Accuracy, AUC score, Recall, and F1 score. Furthermore, the performance of these models is visualized through the illustration of the ROC curve.

### Prevalence of MetS

In our study, we found that the prevalence of MetS was 27.8% among the 8185 participants. Notably, there were significant variations between males and females, with males accounting for 41.7% of the MetS group and females comprising 58.3%. Fig 2 further highlights the dominance of the female population in both the MetS group and the Normal group.

Table 2 displays the overall attributes of the participants in relation to the Mets group and Normal group. The Chi-Square test was employed in SPSS version 28.0 to analyze the data.

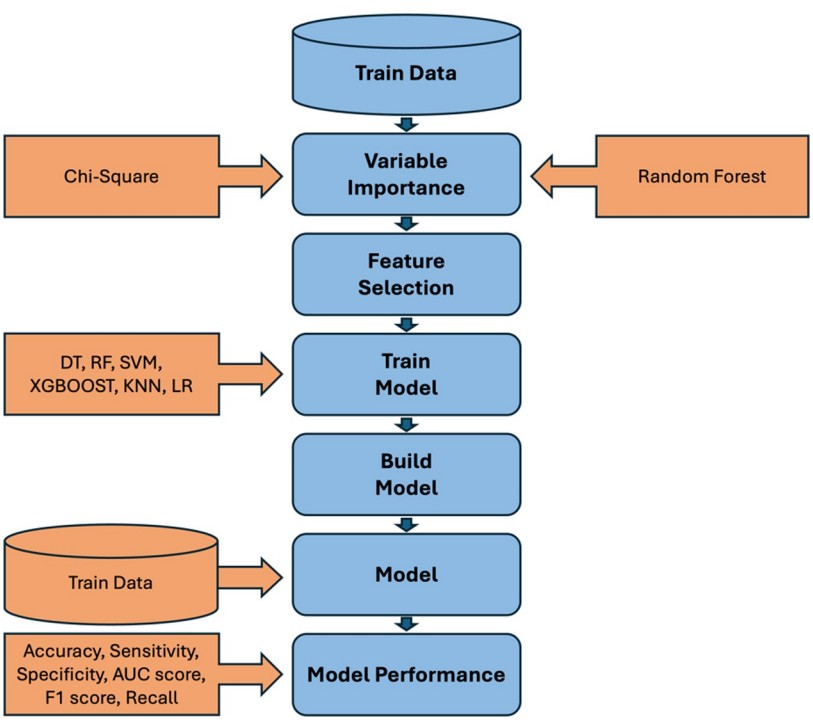

**Fig 1. Diagram to show the process of building the model.**

## Variable importance and key risk factors of metabolic syndrome

In the initial stage of our analysis, we classified our continuous variables, including WHtR, SBP, DBP, WC, HC, FBG, and TGs, into two categories: Yes and No. The classification of continuous variables is crucial for capturing non-linear relationships in models. Categorizing

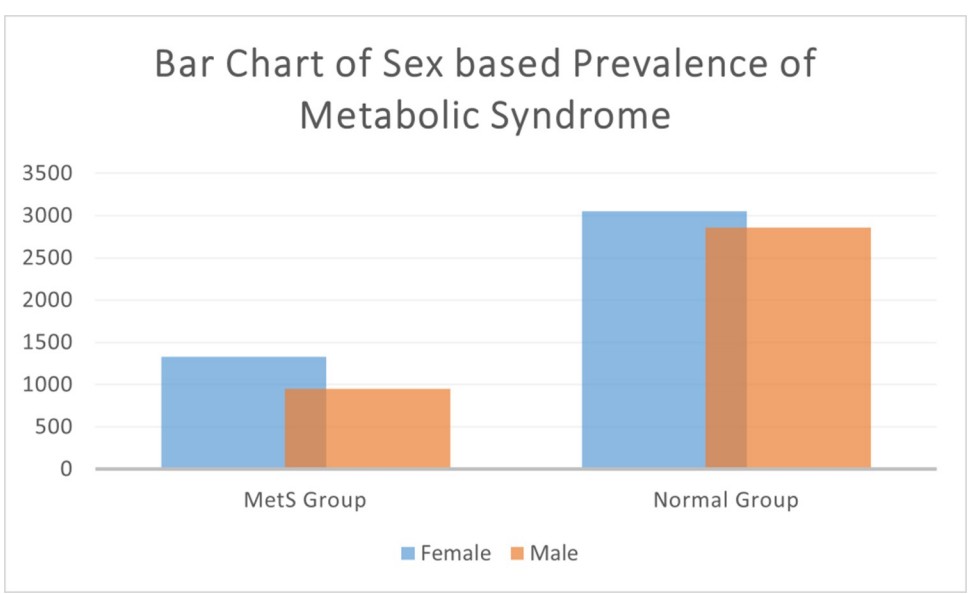

**Fig 2. Bar chart of showing prevalence of MetS regarding sex.**

**Table 2. General descriptions of the participants regarding MetS group and normal group using chi square test.**

| Variables | MetS Group (n = 2275) Count (%) | Normal Group (n = 5910) Count (%) | Total (n = 8185) Count (%) | P-value |
|---|---|---|---|---|
| **Sex** | | | | 0.000 |
| Female | 1327 (58.3) | 3054 (51.7) | 4381 (53.5) | |
| Male | 948 (41.7) | 2856 (48.3) | 3804 (46.5) | |
| **Age** | | | | 0.000 |
| 18–24 | 92 (4.0) | 934 (15.8) | 1026 (12.5) | |
| 25–39 | 723 (31.8) | 2766 (46.8) | 3489 (42.6) | |
| 40–54 | 924 (40.6) | 1579 (26.7) | 2503 (30.6) | |
| 55–69 | 536 (23.6) | 631 (10.7) | 1167 (14.3) | |
| **Residence** | | | | 0.000 |
| Urban | 1239 (54.5) | 2763 (46.8) | 4002 (48.9) | |
| Rural | 1036 (45.5) | 3147 (53.2) | 4183 (51.1) | |
| **Marital Status** | | | | 0.000 |
| Never Married | 41 (1.8) | 449 (8.4) | 535 (6.5) | |
| Currently Married | 2062 (90.6) | 5188 (87.8) | 7250 (88.6) | |
| Separated | 11 (0.5) | 27 (0.5) | 38 (0.5) | |
| Divorced | 9 (0.4) | 20 (0.3) | 29 (0.4) | |
| Widowed | 152 (6.7) | 181 (3.1) | 333 (4.1) | |
| **Smoking** | | | | 0.000 |
| Yes | 411 (18.1) | 1512 (25.6) | 1923 (23.5) | |
| No | 1864 (81.9) | 4398 (74.4) | 6262 (76.5) | |
| **Alcohol** | | | | 0.011 |
| Yes | 149 (6.5) | 477 (8.1) | 626 (7.6) | |
| No | 2126 (93.5) | 5433 (91.9) | 7559 (92.4) | |
| **Eating Extra Salt** | | | | 0.000 |
| Yes | 933 (41) | 2713 (45.9) | 3646 (44.5) | |
| No | 1442 (59) | 3197 (54.1) | 4539 (55.5) | |
| **SBP at Risk** | | | | 0.000 |
| Yes | 1133 (49.8) | 1361 (23.0) | 2494 (30.5) | |
| No | 1142 (50.2) | 4549 (77.0) | 5691 (69.5) | |
| **DBP at Risk** | | | | 0.000 |
| Yes | 1152 (50.6) | 1724 (29.2) | 2876 (35.1) | |
| No | 1123 (49.4) | 4186 (70.8) | 5309 (64.9) | |
| **BMI at Risk** | | | | 0.000 |
| Yes | 252 (11.1) | 271 (4.6) | 523 (6.4) | |
| No | 2023 (88.9) | 5639 (95.4) | (93.6) | |
| **WC at Risk** | | | | 0.000 |
| Yes | 888 (39.0) | 1065 (18.0) | 1953 (23.9) | |
| No | 1387 (61.0) | 4845 (82.0) | 6232 (76.1) | |
| **HC at Risk** | | | | 0.000 |
| Yes | 1082 (47.6) | 1806 (30.6) | 2888 (35.3) | |
| No | 1193 (52.4) | 4104 (69.4) | 5297 (64.7) | |
| **WHtR at Risk** | | | | 0.000 |
| Yes | 1358 (59.7) | 1988 (33.6) | 3346 (40.9) | |
| No | 917 (40.3) | 3922 (66.4) | 4839 (59.1) | |
| **FBG at Risk** | | | | 0.000 |
| Yes | 635 (27.9) | 712 (12.0) | 1347 (16.5) | |
| No | 1640 (72.1) | 5198 (88.0) | 6838 (83.5) | |

(*Continued*)

**Table 2.** (Continued)

| Variables | MetS Group (n = 2275) Count (%) | Normal Group (n = 5910) Count (%) | Total (n = 8185) Count (%) | P-value |
|---|---|---|---|---|
| **TGs at Risk** | | | | 0.000 |
| Yes | 1075 (47.3) | 2011 (34.0) | 3086 (37.7) | |
| No | 1200 (52.7) | 3899 (66.0) | 5099 (62.3) | |
| **HDL at Risk** | | | | 0.036 |
| Yes | 1403 (61.7) | 3495 (59.1) | 4898 (59.8) | |
| No | 872 (38.3) | 2415 (40.9) | 3287 (40.2) | |

SBP = Systolic Blood Pressure, DBP = Diastolic Blood Pressure, BMI = Body Mass Index, WC = Waist Circumference, HP = Hip Circumference, WHtR = Waist and Height Ratio, FBG = Fasting Blood Glucose, TGs = Triglycerides, HDL = High Density Lipoprotein.

variables into ranges helps the model capture complex patterns effectively. This process optimizes algorithm performance by simplifying data representation, making it easier for the algorithm to learn and predict. Standardizing input data format and representing all variable types appropriately is essential for dealing with heterogeneous data. Categorizing continuous variables also improves the interpretability of the model's output, making predictions easier to understand for users [38]. This categorization was based on the ATP III criteria from the original dataset [26]. We made this categorization for our convenience in applying various classification algorithms.

Upon examining Table 3 and Fig 3, we observed that Age, WHtR, SBP, DBP, WC, HC, FBG, Marital Status, TGs, and Residence were identified as key risk factors for Metabolic Syndrome (MetS) through the application of Chi-Square analysis. Among these variables, age had the highest score, followed by WHtR. SBP ranked third in terms of significance. DBP, WC, HC, and FBG held a moderate level of importance. Marital Status, TGs, and Residence were found to have the lowest significance.

After ranking these variables, it became evident that age played a crucial role in the development of Metabolic Syndrome. We observed that individuals above the age of 40 were more susceptible to Metabolic Syndrome diseases. Additionally, WHtR emerged as another prominent risk factor influencing MetS.

In Fig 4, the application of Random Forest on our training dataset reveals that Age, SBP, WHtR, FBG, WC, DBP, marital status, HC, TGs, and smoking are identified as the key risk factors for Metabolic Syndrome (MetS). The Age variable demonstrates the highest score, as confirmed by Chi Square analysis, followed by SBP. WHtR ranks third in importance. FBG,

**Table 3. Top 10 important variables obtained by chi square.**

| Ranking | Variables | Scores |
|---|---|---|
| 01 | Age | 296.402 |
| 02 | WHtR | 188.615 |
| 03 | SBP | 169.351 |
| 04 | DBP | 116.702 |
| 05 | WC | 95.258 |
| 06 | HC | 73.374 |
| 07 | FBG | 49.488 |
| 08 | Marital | 47.375 |
| 09 | TGs | 46.123 |
| 10 | Residence | 19.108 |

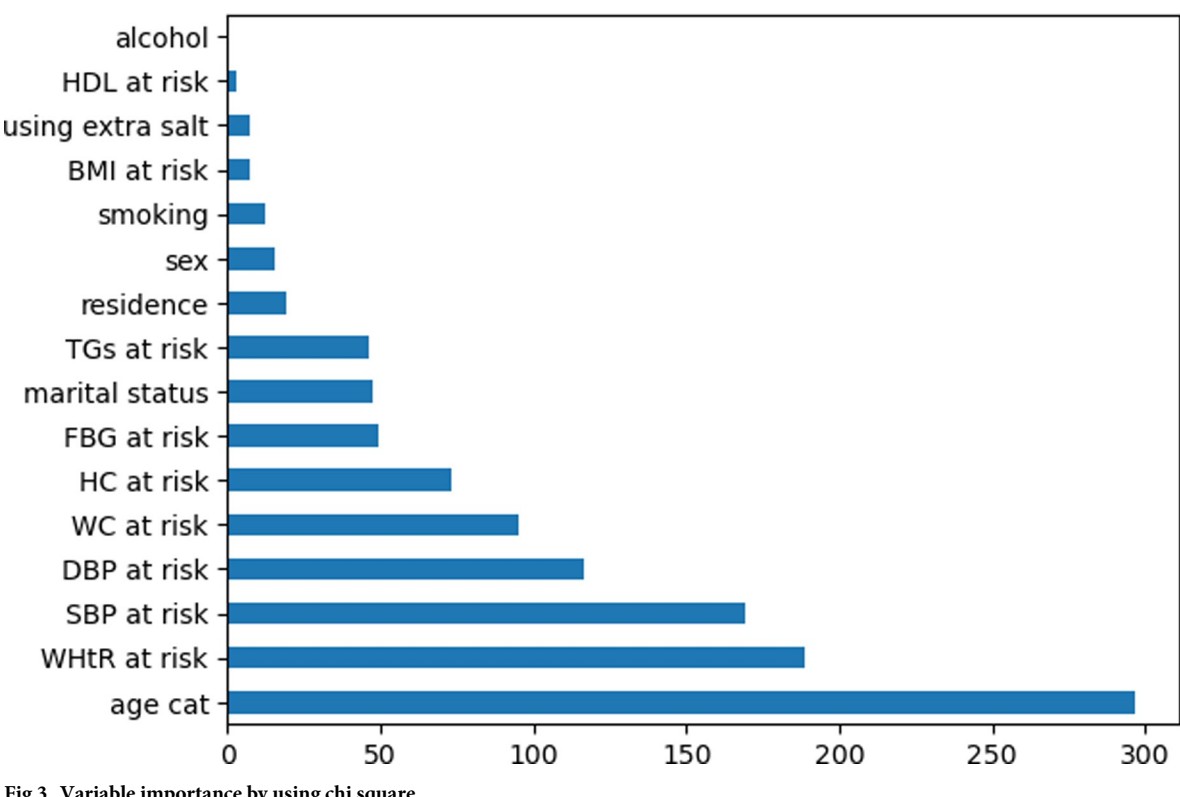

**Fig 3. Variable importance by using chi square.**

WC, DBP, and marital status hold a medium position in terms of significance. On the other hand, HC, TGs, and smoking are considered the least influential factors. The ranking of these factors, similar to Chi-Square analysis, highlights Age as the most crucial determinant for Metabolic Syndrome, followed by Systolic Blood Pressure. It has been established that individuals with elevated systolic blood pressure face a greater risk of developing Metabolic Syndrome.

## Model performance and comparison

The predominant key risk factors of Metabolic Syndrome have been identified as Age, SBP, WHtR, FBG, WC, DBP, HC, and TGs through the Random Forest Variable Importance Measures (VIMs) technique. In order to further investigate these variables, we employed six different ML algorithms, namely Decision Tree, Random Forest, Support Vector Machine, Extreme Gradient Boosting, K-Nearest Neighbors, and Logistic Regression. These algorithms were chosen to explore the potential relationship between the aforementioned eight variables and Metabolic Syndrome.

We evaluate our models through both cross validation and external validation, and we find that our models demonstrate excellent performance. We optimize the parameters for each model through hyperparameter tuning and incorporate them into the model. We mention these parameters in the Table 4.

Table 4 illustrates that the Support Vector Machine (SVM) achieves the highest precision compared to other algorithms, with an accuracy rate of 78%. The SVM accurately identifies 78% of the relevant items. On the other hand, Logistic regression and XGBoost exhibit the highest balanced accuracy (71%) and sensitivity score (63%) among the other algorithms. However, there is only a 63% probability that the model will correctly detect positive cases of

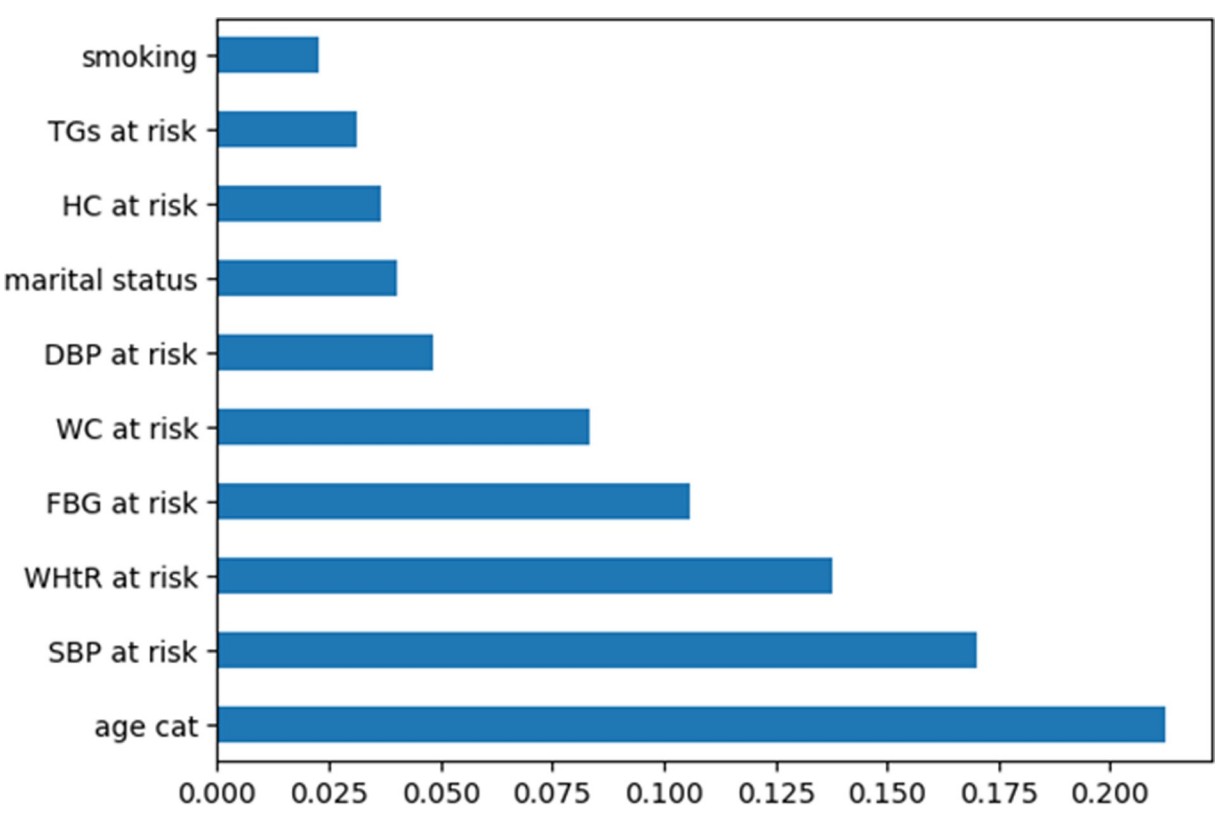

**Fig 4. Variable importance by using random forest.**

Metabolic Syndrome, which is a significantly lower score. In terms of specificity score, XGBoost and KNN outperform other algorithms, with a rate of 79% probability that these models will correctly reject negative cases. Logistic Regression demonstrates the best results in the AUC score, indicating that 75% of items are correctly classified by this algorithm. In terms of Recall, all classifiers exhibit similarly good performance, capturing 75% to 77% of positive items. The F1 score reveals that all classifiers perform moderately well. Overall, considering all metrics, Logistic Regression emerges as the best classifier among the other algorithms.

Table 5 displays the results of the six models with excluded variables such as BMI, HDL, sex, smoking, alcohol, and using extra salt. The presence of these variables has been found to

**Table 4. The classification performance results of the models.**

| Classifier | Precision | Balanced Accuracy | Sensitivity | Specificity | AUC | Recall | F1-Score |
|---|---|---|---|---|---|---|---|
| **Decision Tree** (criterion = 'Gini', max depth = 4, max leaf nodes = 10, min samples leaf = 0.05, min samples split = 2) | 0.75 | 0.68 | 0.58 | 0.77 | 0.73 | 0.75 | 0.71 |
| **Random Forest** (max depth = 8, no of estimators = 40) | 0.76 | 0.69 | 0.61 | 0.78 | 0.74 | 0.76 | 0.73 |
| **Support Vector Machine**(C = 1, gamma = 0.1, kernel = 'rbf') | 0.78 | 0.70 | 0.62 | 0.78 | 0.68 | 0.76 | 0.72 |
| **Extreme Gradient Boosting** (learningrate = 0.1,maxdepth = 2,no of estimators = 180 | 0.74 | 0.71 | 0.63 | 0.79 | 0.70 | 0.77 | 0.73 |
| **K-Nearest Neighbors** (no of estimators = 19) | 0.75 | 0.68 | 0.57 | 0.79 | 0.72 | 0.75 | 0.73 |
| **Logistic Regression**(C = 1, solver = liblinear) | 0.77 | 0.71 | 0.63 | 0.78 | 0.75 | 0.77 | 0.74 |

**Table 5. The classification performance results of the models considering excluded variables.**

| Classifier | Precision | Balanced Accuracy | Sensitivity | Specificity | AUC | Recall | F1-Score |
|---|---|---|---|---|---|---|---|
| **Decision Tree** (criterion = 'Gini', max depth = 4, max leaf nodes = 10, min samples leaf = 0.05, min samples split = 2) | 0.73 | 0.36 | 0 | 0.73 | 0.36 | 0 | 0 |
| **Random Forest** (max depth = 8, no of estimators = 40) | 0.73 | 0.86 | 1 | 0.73 | 0.86 | 1 | 0.84 |
| **Support Vector Machine** (C = 1, gamma = 0.1, kernel = 'rbf') | 0.73 | 0.36 | 0 | 0.73 | 0.36 | 0 | 0 |
| **Extreme Gradient Boosting** (learning rate = 0.1, max depth = 2, no of estimators = 180 | 0.72 | 0.56 | 0.39 | 0.73 | 0.56 | 0.39 | 0.50 |
| **K-Nearest Neighbors** (no of estimators = 19) | 0.72 | 0.58 | 0.43 | 0.74 | 0.58 | 0.43 | 0.53 |
| **Logistic Regression** (C = 1, solver = liblinear) | 0.73 | 0.61 | 0.48 | 0.73 | 0.61 | 0.48 | 0.57 |

diminish the overall performance of the models. Consequently, we have opted to remove these variables from our analysis due to their lack of significance in relation to our results.

Fig 5 illustrates the identical concept that was previously discussed. Logistic Regression stands out as the most effective classification algorithm when compared to other algorithms in the field. Based on the findings presented in Fig 5, XGBoost can be regarded as the poorest performing classification algorithm, exhibiting the lowest precision in comparison to other algorithms in the same category.

Fig 6 illustrates the ROC curve, which depicts the performance of different classification algorithms. According to the results, Logistic Regression emerges as the most effective classification algorithm, while SVM is identified as the least accurate classification algorithm.

## Discussion and recommendation

Our research represents a pioneering effort in predicting potential risk factors of MetS using MLTs. The study encompassed a total of 8185 participants from various regions of Bangladesh and incorporated 16 distinct variables. These variables encompassed anthropometric data, lifestyle-related features gathered through questionnaires, and biochemical test results which were collected from "National STEPS Survey for Non-communicable Diseases Risk Factors in Bangladesh 2018". Upon conducting the required computations, our findings revealed that 27.8% of the population exhibited a prevalence of MetS [39, 40].

Notably, there were remarkable disparities between sexes, with males comprising 41.7% of the MetS group and females accounting for 58.3% [41, 42]. The higher prevalence of MetS among females can be attributed to various factors, including sociocultural activities, psychosocial behaviors, socioeconomic status, genetic inheritance, and hormonal changes. These factors make females more susceptible to developing MetS compared to males [43].

In this research, a collection of health parameters was prioritized using Chi-Square and then compared to Random Forest to determine the significance of each variable. The findings revealed that the primary predominant variables for MetS in our sample of the Bangladeshi population based on their importance were Age, WHtR, SBP, DBP, WC, FBG, HC, and TGs. Interestingly, six out of these eight variables align with the criteria proposed by ATP III for classifying individuals with MetS [44].

It is worth noting that WHtR was ranked as the second variable in terms of significance, which is a noteworthy discovery, particularly concerning the obesity crisis in our nation and its association with cardiovascular diseases, the leading cause of illness and death globally and in Bangladesh. Abdominal obesity has emerged as an indicator of cardiometabolic risk, prompting considerable endeavors to identify a suitable anthropometric measurement that

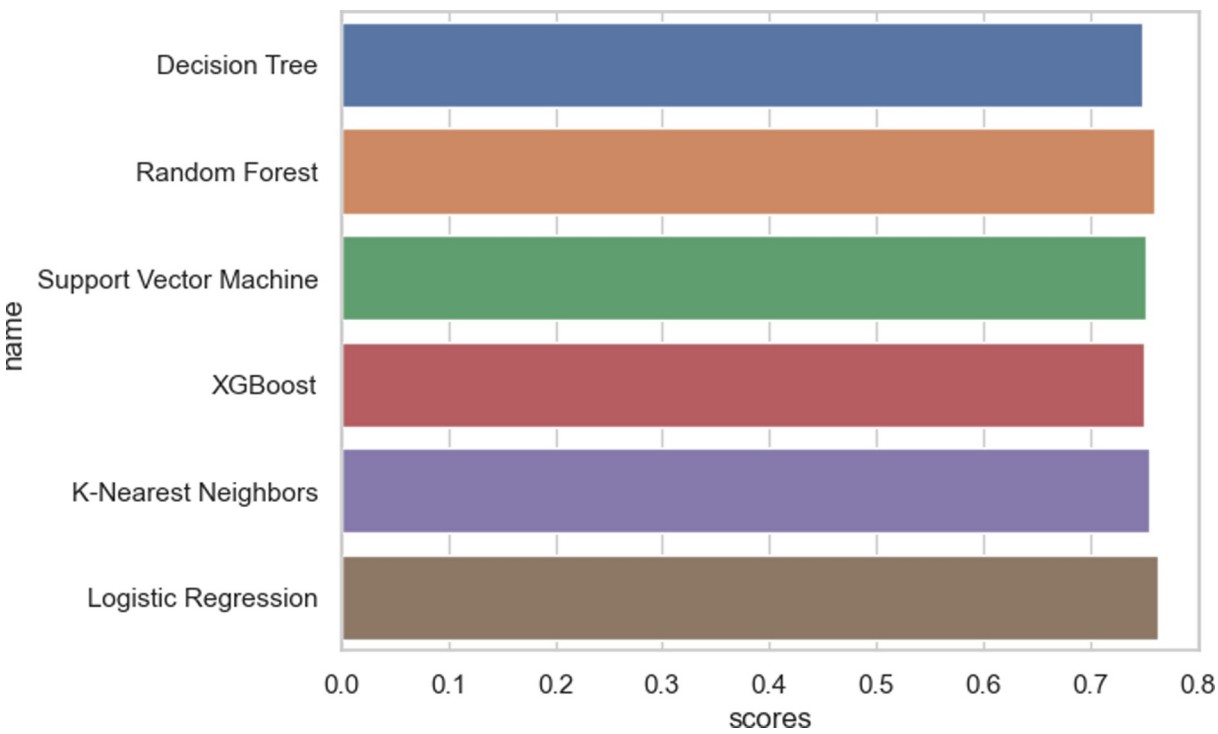

**Fig 5. Bar chart of displaying model performance.**

accurately reflects the accumulation of fat in the abdominal region and can be conveniently obtained without the need for advanced technological equipment [17, 45].

It is also worth noting that anthropometric indexes are significantly impacted by various factors such as age, sex, and ethnicity. Consequently, selecting a suitable index can be a daunting endeavor. When utilizing Chi-Square and Random Forest in our research, age emerges as the most crucial factor. Individuals who are older than 40 years are found to be in the high-risk category for MetS. While BMI has been widely utilized as a measure of body fat content, in our research it fails to accurately reflect abdominal obesity [46, 47].

A simple and effective method for assessing abdominal obesity and associated metabolic risk is WC measurement. It is worth noting that in our research WC is the fifth significant feature as a risk factor for Mets and abdominal obesity significantly influences the emergence of MetS.

The Random Forest Variable Important Measures (VIMs) technique in Fig 4 revealed that BMI, HDL, Marital Status, Residence, Sex, Using Extra Salt, and Smoking had lower scores and were therefore not considered in the modeling. The inclusion of these variables resulted in a decrease in the overall accuracy of the models. Conversely, excluding these variables led to an increase in the overall accuracy of the models [26].

Briefly, our research has revealed a moderately higher prevalence of Metabolic Syndrome (MetS) among the Bangladeshi Population, particularly among females. The primary risk factors identified were Age, Waist and Height Ratio (WHtR), High Systolic and Diastolic Blood Pressure, Waist Circumference (WC), Fasting Blood Glucose (FBG), Hip Circumference (HC), and Triglycerides (TGs), which aligns with the conventional definition of MetS established by various organizations.

Machine learning models have demonstrated potential in clinical settings for early detection and intervention of metabolic syndrome. These models utilize non-invasive factors,

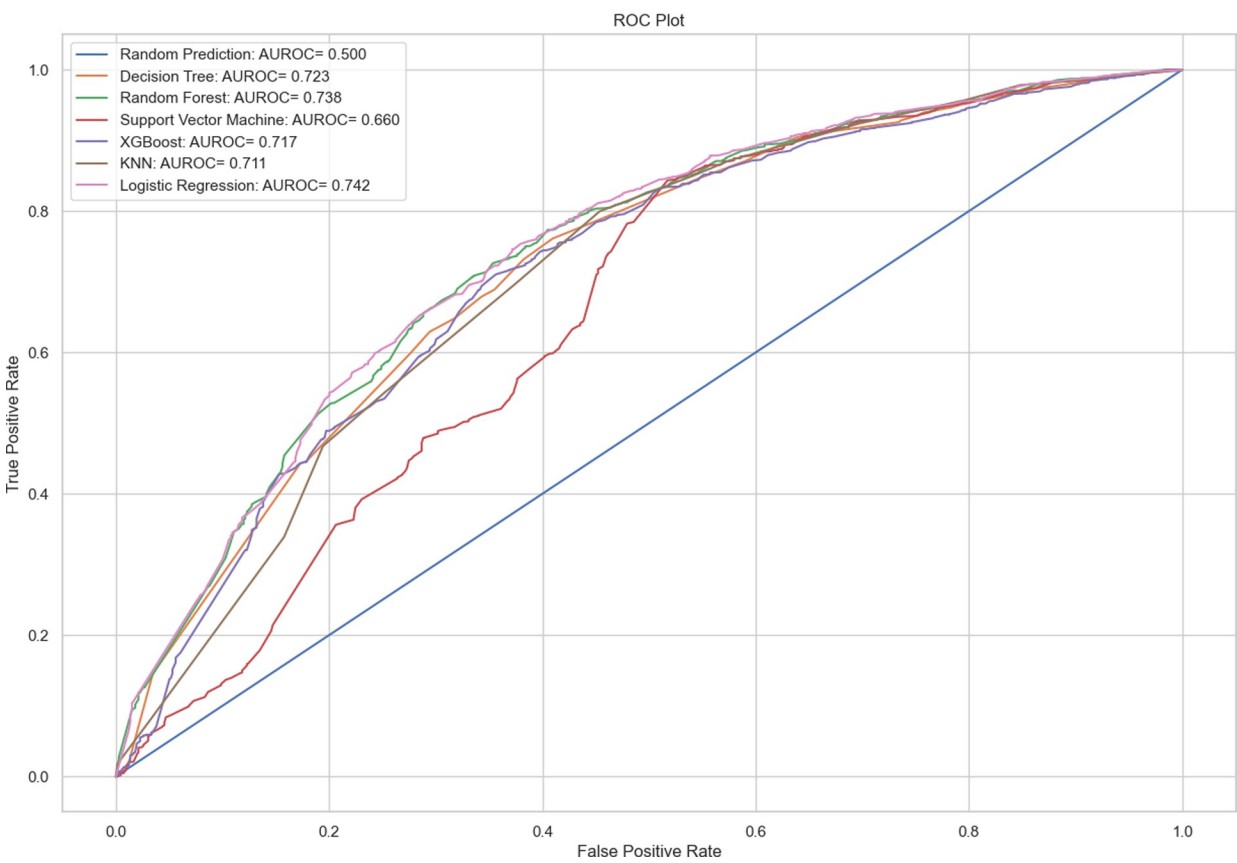

**Fig 6. ROC curve.**

making them a cost-effective option for large-scale screening. By taking into account variables such as gender, Age, WHtR, SBP, DBP, WC, FBG, HC, and TGs, these models can pinpoint individuals at high risk. Additionally, they can analyze health records, lifestyle factors, and medical history to provide personalized risk assessments. With a focus on accuracy, these models allow for timely intervention and customized treatment plans for individuals at risk of metabolic syndrome [48, 49]. Future investigation is necessary to further explore the integration of machine learning models into healthcare systems in Bangladesh for early detection and intervention strategies.

## Recommendations

It is recommended that expanding on the importance of giving higher priority to older individuals in preventing Metabolic Syndrome (MetS), it is essential to recognize that this population is more susceptible to developing MetS due to age-related physiological changes and lifestyle factors. By prioritizing older individuals, healthcare centers can focus on providing targeted interventions and preventive measures to reduce the risk of MetS. Furthermore, it is crucial to educate females about the risk factors associated with MetS, as they are also more prone to developing this condition. Women often experience hormonal changes throughout their lives, such as during pregnancy and menopause, which can contribute to the development of MetS. By raising awareness among females, healthcare centers can empower them to make informed decisions regarding their health and take necessary steps to prevent MetS [50].

Overall, prioritizing older individuals and providing targeted education on MetS risk factors, with a special emphasis on WHtR, can significantly contribute to the prevention and management of this condition. By promoting awareness and encouraging individuals to be conscious of their weight, WC, and HC, healthcare centers can empower individuals to take control of their health and reduce the burden of MetS in the population [50, 51].

## Conclusion

To summarize, our research successfully estimated the prevalence of MetS in Bangladesh, which was found to be 27.8%. Notably, the female population showed a higher prevalence of MetS. Through the implementation of Random Forest and Chi-Square methods, we identified several potential risk factors for MetS, including increased Age, WHtR, SBP, DBP, WC, FBG, HC, and TGs.These models can be utilized to assess the influencing factors of various diseases and contribute to treatment strategies and decision-making processes.

Furthermore, it is crucial to recognize MetSasa global epidemic. In order to limit its further spread and reduce associated morbidity and mortality, it is imperative to implement regional measures that prioritize primary prevention.

## Limitations and strength

**Limitations.** There is a limitation that necessitates attention in our study.

**Exclusion of potentially relevant variables.** Certain variables were excluded from the analysis due to their insignificant results. Moreover, the performance of our model is not particularly high. Additionally, the significance of BMI, a crucial factor for abdominal obesity, was not observed in our research. Lastly, HDL, an important factor for cardiovascular disease, also exhibited insignificance in our study [52].

## Strength

**Strengths of our study.** Despite these limitations, the primary strength of this study lies in the extensive size of the dataset, which was obtained from the "National STEPS Survey for Non-communicable Diseases Risk Factors in Bangladesh 2018" survey. We employed six significant classification algorithms to predict our results. Notably, our study is the pioneering endeavor of its kind to forecast the prevalence and potential risk factors of MetS in Bangladesh.

## Supporting information

**S1 Data.**
(ZIP)

## Acknowledgments

We would like to express our sincere appreciation for obtaining the dataset from the "National STEPS Survey for Non-communicable Diseases Risk Factors in Bangladesh 2018".

## Author Contributions

**Conceptualization:** Md Farhad Hossain.

**Data curation:** Mst. Nira Akter, Ainur Nahar.

**Formal analysis:** Shaheed Hossain.

**Investigation:** Md Farhad Hossain.

**Methodology:** Md Farhad Hossain, Shaheed Hossain.

**Project administration:** Ainur Nahar.

**Software:** Shaheed Hossain.

**Supervision:** Bowen Liu.

**Validation:** Md Farhad Hossain, Bowen Liu.

**Visualization:** Mst. Nira Akter, Md Omar Faruque.

**Writing – original draft:** Md Farhad Hossain, Shaheed Hossain.

**Writing – review & editing:** Bowen Liu, Md Omar Faruque.

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
