## [Decision Letter · Decision Letter 0]

8 Mar 2024

PONE-D-24-05709

Metabolic Syndrome Predictive Modeling in Bangladesh applying Machine Learning Approach

PLOS ONE

Dear Dr. Hossain,

Thank you for submitting your manuscript to PLOS ONE. After careful consideration, we feel that it has merit but does not fully meet PLOS ONE’s publication criteria as it currently stands. Therefore, we invite you to submit a revised version of the manuscript that addresses the points raised during the review process.

We look forward to receiving your revised manuscript.

Kind regards,

Donovan Anthony McGrowder, PhD., MA., MSc

Academic Editor

PLOS ONE

Journal Requirements:

7. We note that Figure 1 in your submission contain map/satellite images which may be copyrighted. All PLOS content is published under the Creative Commons Attribution License (CC BY 4.0), which means that the manuscript, images, and Supporting Information files will be freely available online, and any third party is permitted to access, download, copy, distribute, and use these materials in any way, even commercially, with proper attribution. For these reasons, we cannot publish previously copyrighted maps or satellite images created using proprietary data, such as Google software (Google Maps, Street View, and Earth). For more information, see our copyright guidelines: http://journals.plos.org/plosone/s/licenses-and-copyright.

Additional Editor Comments:

Dear Dr. Hossain,

Your manuscript “Metabolic Syndrome Predictive Modeling in Bangladesh applying Machine Learning Approach” has been assessed by our reviewers. They have raised a number of points which we believe would improve the manuscript and may allow a revised version to be published in PLOS ONE. Their reports, together with any other comments given are below.

If you are able to fully address these points, we would encourage you to submit a revised manuscript to PLOS ONE by the date given below.

 Best regards,

Dr. Donovan McGrowder

Reviewers' comments:

Reviewer's Responses to Questions

**Comments to the Author**

1. Is the manuscript technically sound, and do the data support the conclusions?

Reviewer #1: Yes

Reviewer #2: Yes

Reviewer #3: Yes

2. Has the statistical analysis been performed appropriately and rigorously? 

Reviewer #1: Yes

Reviewer #2: Yes

Reviewer #3: Yes

3. Have the authors made all data underlying the findings in their manuscript fully available?

Reviewer #1: Yes

Reviewer #2: Yes

Reviewer #3: Yes

4. Is the manuscript presented in an intelligible fashion and written in standard English?

Reviewer #1: Yes

Reviewer #2: Yes

Reviewer #3: Yes

5. Review Comments to the Author

Reviewer #1: Abstract:

- The abstract should clearly state the main objectives, methodology, key results, conclusions and limitations of the study. Currently it only summarizes the background.

- Include the key risk factors identified, prevalence of MetS found, and main ML model utilized in the results.

Introduction:

- Expand more on the rationale for identifying risk factors and predicting MetS prevalence in the Bangladeshi population specifically.

- after ''and working in irregular shifts[7].'' discuss the role in particular of obstructive sleep apnea. cite doi:10.3390/life13030702.

- Provide more background on MetS diagnostic criteria and associated health risks to establish significance.

Methods:

- Describe the National STEPS Survey sampling methodology, data collection procedures, and variables obtained in more detail.

- Explain how MetS was defined - which diagnostic criteria used.

- Specify why certain variables were categorized and encoding approaches for ML models.

- State specific ML models, parameters, evaluation metrics, and statistical analysis done.

Results:

- Focus this section only on key results - prevalence, risk factors, model performance. Remove general discussion.

- Include quantitative performance metrics for each ML model - precision, recall, ROC, etc.

Discussion:

- Limitations should note small subset of survey data used, exclusion of potentially relevant variables like lipids. cite doi:10.1016/j.compbiomed.

- Discussion of inflammation markers and future studies seems speculative without any results presented. Would remove.

- Avoid overstating conclusions on predictive capabilities of models until externally validated.

Reviewer #2: Dear authors,

I have now completed the review of the manuscript titled "Metabolic Syndrome Predictive Modeling in Bangladesh applying Machine Learning Approach".

The author aims to predict the prevalence of Metabolic Syndrome (MetS) in Bangladesh using various machine learning algorithms. The study analyzes data from 8185 participants, identifying key risk factors for MetS and assessing the performance of different models in predicting the condition.

The manuscript is interesting and, in general, fairly well-written.

I have some suggestions to further improve the quality of the manuscript.

I would like to suggest that the authors address these limitations in the article, either by discussing them in the limitations section or, where feasible, by making the appropriate revisions:

1. It's crucial to assess how well the models generalize to unseen data. Cross-validation techniques and external validation on datasets from different populations could enhance the robustness of the findings.

2. The study could benefit from a direct comparison with existing predictive models for MetS to highlight its contributions or improvements.

3. While the study employs feature selection techniques, further exploration into dimensionality reduction methods could improve model performance and interpretability.

4. The exclusion of certain variables deemed insignificant could potentially overlook complex interactions between features that contribute to MetS. A more detailed analysis or justification for the exclusion of these variables might provide deeper insights.

5. The practical applicability of the models in clinical settings is not extensively discussed. Future work could focus on how these models can be integrated into healthcare systems for early detection and intervention strategies.

6. Mention breifly deep learning methods article and add in the reference. For example: "An Adaptive Ensemble Deep Learning Framework for Reliable Detection of Pandemic Patients", "Deep Learning Network Selection and Optimized Information Fusion for Enhanced COVID-19 Detection". Deep learning could offer insight into the future research direction.

Thank you for your valuable contributions to our field of research. I look forward to receiving the revised manuscript.

Reviewer #3: In this study, Hossain et al. perform machine learning to identify risk factors for metabolic syndrome. The topic is interesting, and data support the authors’ conclusions. The story is clear; however, the method section is incomplete. The authors write general information of procedures, such as what sensitivity is or how specificity is calculated. This basic and general information is not necessary. The authors should write procedures the authors performed in this study, but no information is provided in this manuscript, which is not appropriate. The authors should provide more detailed information of procedures, such as R and its version, packages, parameters used in analysis, etc.

6. PLOS authors have the option to publish the peer review history of their article (what does this mean?). If published, this will include your full peer review and any attached files.

Reviewer #1: **Yes: **Antonino

Reviewer #2: No

Reviewer #3: No

---

## [Author Response · Author response to Decision Letter 0]

26 Apr 2024

Reviewer 1:

Reviewer comments Responses

The abstract should clearly state the main objectives, methodology, key results, conclusions, and limitations of the study. Currently it only summarizes the background. We appreciate your feedback. The primary goals, methodology, important findings, conclusions, and limitations of the study have all been spelled out in detail in the abstract.

Include the key risk factors identified, prevalence of MetS found, and main ML model utilized in the results. Thank you for your recommendations. The major machine learning model used in the results, the prevalence of MetS, and the key risk variables revealed have all been presented in the "abstract".

Expand more on the rationale for identifying risk factors and predicting MetS prevalence in the Bangladeshi population specifically. We appreciate your recommendations. The reasoning behind determining risk variables and forecasting the incidence of MetS in the Bangladeshi population in particular is covered in greater detail in the "Introduction section" of our publication.

after ‘and working in irregular shifts [7].'discuss the role in particular of obstructive sleep apnea. cite doi:10.3390/life13030702. Thanks for your suggestions. According to this research doi:10.3390/life13030702, we have addressed the role of obstructive sleep apnea specifically in the "Introduction section." 

Provide more background on MetS diagnostic criteria and associated health risks to establish significance. We appreciate your recommendations. To establish significance, we have included further background information on MetS diagnostic criteria and related health hazards in the "Introduction section" of our publication.

Describe the National STEPS Survey sampling methodology, data collection procedures, and variables obtained in more detail. Thank you for your recommendations. More information on the National STEPS Survey sampling methodology, data collection techniques, and variables gathered may be found in the section titled "Data Source in the Methodology section."

Explain how MetS was defined - which diagnostic criteria used Thanks for your suggestionsThe definition of MetS and the diagnostic criteria that were applied are covered in the section on "diagnostic criteria in the Methodology section."

Specify why certain variables were categorized and encoding approaches for ML models We have explained why specific variables were categorized and encoding methodologies for ML models in the section "Variable importance and key risk factors of metabolic syndrome in the Results and analysis section."

State specific ML models, parameters, evaluation metrics, and statistical analysis done Thanks for your suggestions. Certain machine learning models, parameters, assessment metrics, and statistical analysis have been specified in the "Results and analysis" portion of Tables 2, 3, and 4.

Focus this section only on key results - prevalence, risk factors, model performance. Remove general discussion We appreciate your recommendations. We only included the most important findings—prevalence, risk variables, and model performance—in the "Results and analysis" section.

Include quantitative performance metrics for each ML model - precision, recall, ROC, etc. Thank you for your tremendous recommendations. We incorporated numerical performance indicators, such as precision, recall, ROC, and so on, for every machine learning model in our manuscript's "Results and analysis section in Table 4."

Limitations should note small subset of survey data used, exclusion of potentially relevant variables like lipids. cite doi:10.1016/j.compbiomed We appreciate your recommendations. We have mentioned in the "Limitations Section" that a tiny subset of survey data was utilized, that potentially relevant factors like lipids were excluded, and that we cited this article doi:10.1016/j.compbiomed.

Discussion of inflammation markers and future studies seems speculative without any results presented. Would remove. We appreciate your insightful recommendations. The part about future research and signs of inflammation has been removed.

Avoid overstating conclusions on predictive capabilities of models until externally validated Thanks for your valuable suggestions. This has been deleted from the "Conclusion section."

Reviewer #2

Reviewer comments Responses

It's crucial to assess how well the models generalize to unseen data. Cross-validation techniques and external validation on datasets from different populations could enhance the robustness of the findings. Your insightful recommendations are appreciated. After examining datasets from various populations, we have conducted cross-validation and external validation on our models and found that they perform well.

The study could benefit from a direct comparison with existing predictive models for MetS to highlight its contributions or improvements We appreciate your insightful recommendations. There are no current MetS prediction models available that we can compare our model to from the standpoint of the Bangladeshi population. Our models showed marginally less accuracy when we compared them to the state-of-the-art models from China and Mexico.

While the study employs feature selection techniques, further exploration into dimensionality reduction methods could improve model performance and interpretability. Thanks for your valuable suggestions. Although we have looked into dimensionality reduction techniques, our model's performance remains unchanged.

The exclusion of certain variables deemed insignificant could potentially overlook complex interactions between features that contribute to MetS. A more detailed analysis or justification for the exclusion of these variables might provide deeper insights. Table 5 in the "Results and analysis section" illustrates how these variables lower the models' overall performance. Because of this, we have decided to eliminate these factors from our study because they don't really matter for our findings. We think more research is necessary to ignore the intricate relationships between the characteristics that cause MetS.

The practical applicability of the models in clinical settings is not extensively discussed. Future work could focus on how these models can be integrated into healthcare systems for early detection and intervention strategies We appreciate your recommendations. The "Discussion and recommendations section" of our publication contains a brief discussion of the models' usefulness in clinical settings. To further understand how machine learning models might be incorporated into Bangladeshi healthcare systems for early detection and intervention measures, we think more research is needed.

Mention briefly deep learning methods article and add in the reference. For example: "An Adaptive Ensemble Deep Learning Framework for Reliable Detection of Pandemic Patients", "Deep Learning Network Selection and Optimized Information Fusion for Enhanced COVID-19 Detection". Deep learning could offer insight into the future research direction. Thanks for your important suggestions. In the "Related work section" of our manuscript, we have included a brief reference of an article on deep learning that may provide some light on potential future study directions.

Reviewer #3

Reviewer comments Responses

This basic and general information is not necessary. The authors should write procedures the authors performed in this study, but no information is provided in this manuscript, which is not appropriate. The authors should provide more detailed information of procedures, such as R and its version, packages, parameters used in analysis, etc. Thanks for your suggestions. We removed the general discussion as possible. We included the author’s contribution in this manuscript. We provided detailed information of procedures. We used SPSS Python instead of R. We included its version, libraries and parameter used in analysis.

---

## [Decision Letter · Decision Letter 1]

13 Aug 2024

Metabolic Syndrome Predictive Modeling in Bangladesh applying Machine Learning Approach

PONE-D-24-05709R1

Dear Dr. Hossain,

We’re pleased to inform you that your manuscript has been judged scientifically suitable for publication and will be formally accepted for publication once it meets all outstanding technical requirements.

Kind regards,

Aleksandra Klisic

Academic Editor

PLOS ONE

Additional Editor Comments (optional):

Reviewers' comments:

Reviewer's Responses to Questions

**Comments to the Author**

1. If the authors have adequately addressed your comments raised in a previous round of review and you feel that this manuscript is now acceptable for publication, you may indicate that here to bypass the “Comments to the Author” section, enter your conflict of interest statement in the “Confidential to Editor” section, and submit your "Accept" recommendation.

Reviewer #1: All comments have been addressed

Reviewer #2: All comments have been addressed

Reviewer #3: All comments have been addressed

2. Is the manuscript technically sound, and do the data support the conclusions?

Reviewer #1: Yes

Reviewer #2: Yes

Reviewer #3: (No Response)

3. Has the statistical analysis been performed appropriately and rigorously? 

Reviewer #1: Yes

Reviewer #2: Yes

Reviewer #3: (No Response)

4. Have the authors made all data underlying the findings in their manuscript fully available?

Reviewer #1: Yes

Reviewer #2: Yes

Reviewer #3: (No Response)

5. Is the manuscript presented in an intelligible fashion and written in standard English?

Reviewer #1: Yes

Reviewer #2: Yes

Reviewer #3: (No Response)

6. Review Comments to the Author

Reviewer #1: The manuscript Is improved and the Authors performed all the revisions required. I can suggest tò accept.

Reviewer #2: All comments were addressed. I declare no further comments to be included. Thank you for considering my opinion.

Reviewer #3: (No Response)

7. PLOS authors have the option to publish the peer review history of their article (what does this mean?). If published, this will include your full peer review and any attached files.

Reviewer #1: **Yes: **None

Reviewer #2: No

Reviewer #3: No
